# Research on the impact of equity incentive model on enterprise performance: A mediating effect analysis based on executive entrepreneurship

**Jinrong Ma**, **Hongbo Wang** *

School of Business Administration, University of Science and Technology Liaoning, Anshan City, China

* whb@ustl.edu.cn

**Data Availability Statement:** All code and data are archived at https://osf.io/m6npy/files/osfstorage.

## Abstract

In implementing the equity incentive system, this paper delves into the listed enterprises' selection of equity incentive models. While previous research has extensively covered the effects, models, and influencing factors of equity incentives, there needs to be more in-depth literature focusing on the diverse incentive models and their impact on corporate performance. Notably, there needs to be more literature on considering entrepreneurial spirit as a mechanism. It aims to explore the relationship between executives' choices under different incentive models, the entrepreneurial spirit fostered by these models, and their combined impact on corporate performance. The findings reveal that adopting the restricted stock incentive model by listed enterprises implementing the equity incentive system significantly positively affects enterprise performance. Mechanistic tests show that when a company implements the restricted stock incentive model, executives prioritize maximizing their interests, leading them to embrace more risk in their investment decisions. This behavior, in turn, stimulates the adventurous spirit of executives, positively impacting enterprise performance, particularly pronounced in companies with more concentrated executive power. Moreover, executives may be more inclined to invest in high-risk, high-reward innovative projects, a behavior indicative of innovation and more prevalent in firms with higher research and development (R&D) investment. However, the limitation of this paper is that the study evaluates the operation of the equity incentive system in China by taking listed companies in China as an example, which is not necessarily suitable for foreign developed capitalist countries. This study contributes to the study of principal-agent problems by exploring the relationship between executives, entrepreneurship and firm performance.

## 1. Introduction

Since the implementation of the equity incentive system in China in 2006, it has gradually become a normalized arrangement for listed companies to solve the principal-agent problem, and the relationship between equity incentive and enterprise performance has also become a

**Funding:** This research was funded by Natural Science Foundation of China [No. 71771112], and Project of Liaoning Provincial Federation Social Science Circles of China [No. L20BGL047, L16BJY011]. The funders had no role in study design, data collection and analysis, decision to publish, or preparation of the manuscript.

**Competing interests:** The authors have declared that no competing interests exist.

hot issue for scholars to study [1]. Most scholars believe that the establishment of incentive and constraint mechanisms can solve the principal-agent problem, so it is crucial that the core contract of the equity incentive system is set up in a way that establishes incentives and at the same time brings constraints to the incentive recipients. When setting up the elements of equity incentive contracts in listed companies, the choice of incentive model is the primary issue considered by both parties of interest. Compared with the more diversified incentive models abroad, domestic enterprises mainly focus on two models of stock option incentive and restricted stock incentive regarding the selection of equity incentive models, and most of the existing equity incentives for executives are based on executive shareholding [2, 3]. As the organizer of a listed company, executives' decisions basically determine the effectiveness of the company and even affect the survival of the company. Several studies have shown that entrepreneurship of executives positively affects enterprise performance in business organization activities [4–6]. Entrepreneurship includes the ability to innovate, create value, take risks and pursue excellence. This ability helps executives identify, seize and capitalize on business opportunities and gives companies an edge over their competition. Therefore, it is significant to explore whether different equity incentive models will stimulate executives' entrepreneurship and thus affect enterprise performance.

In summary, the innovation of this paper is to consider the choices made by executives of Chinese listed companies under different incentive models and the entrepreneurial spirit inspired by such incentive models, i.e., the spirit of adventure and innovation, and their impact on enterprise performance. Most of the current literature has used traditional foreign time-based equity incentives as the subject of study. However, the setting of performance evaluation indexes in the equity incentive programs of listed companies is a mandatory target of the relevant regulatory authorities in China, which leads to the difference between the development of equity incentives in China and foreign countries. So, in the context of performance-based equity incentives in China, how do executives choose between different incentive models? What other factors besides agency issues affect the choice of incentive model? How do the effects of different incentive models differ in practice? In the current context of deepening capital market reform, it is of great theoretical and practical significance to explore the above issues to evaluate the practical effect of China's equity incentive system and improve the function of capital market to serve economic transformation and upgrading.

Most existing literature considers the foreign capital market as the research object. Since the equity incentive system in China's capital market started late, there are fewer related studies in China. Moreover, the existing literature on executives and corporate performance is mostly about exploring the relationship between executive shareholding and corporate performance, and a few literatures have explored the intensity of equity incentives and corporate performance. There is a particular gap in the literature related to the impact of the equity incentive model on corporate performance, and there is a gap in the literature that explores the mediating mechanism between the two.

This study first uses the method of mathematical analysis to compare the benefits of different incentive models from the perspective of executives and uses this as a hypothesis for empirical analysis. The data of listed companies in China A-share market from 2010 to 2020 are used as the sample, and a two-way panel fixed-effects regression is conducted with the model of equity incentives implemented by the sample companies as the explanatory variables to analyze the impact of the model of equity incentives on enterprise performance and the mediating role of entrepreneurship of executives in between. Robust regressions were also performed using GMM, replacement variables and Sobel test. Finally, we summarize the empirical results and point out the shortcomings and future development directions of this study, so as to

provide theoretical reference for different types of listed companies in choosing the equity incentive model.

The structure of the paper is organized as follows: The first part is the Introduction, which describes the significance and originality of the study. The second part is the Literature Review, which shows the previous research methods and results and proposes the research method for this paper. The third part is the Theoretical Analysis and Research Hypothesis, which is based on the theoretical analysis and puts forward the research hypothesis of this paper. The fourth part is the Research Design, which introduces the data sources of the model, defines the variables and constructs the empirical model of this paper. The fifth part is the Empirical Results and Analysis; the results of the empirical evidence are reported. The sixth part is the Discussion and Conclusion, which discusses the above experimental results, concludes this paper, and points out the shortcomings and future development direction of this study to provide theoretical references for different types of listed companies in choosing equity incentive models.

## 2. Literature review

The early foreign literature on research related to equity incentive models can be divided into two categories: the first category focuses on the effect of agency costs on the choice of incentive model, and this category is more numerous. The second category explores the influence of corporate profits, taxes, and dividend distributions on the choice of incentive model. According to principal-agent theory, the first type of literature argues that firms prefer the more incentive-based stock options as the equity incentive model in order to alleviate agency problems. According to numerous studies, stock options, as an equity incentive model, can better motivate incentive recipients to risk investing for higher returns. In contrast, restricted stock is an equity incentive that is granted free of charge to the incentive recipients and is less motivating [7–9]. Compared to the first type of literature, the second type of literature is smaller and has a more fragmented perspective. Core and Guay [10] argued that stock options had significant advantages for embellishing a company's profit statement. Dechow, et al [11] found that companies with high tax rates pay higher taxes if they embellish their statements to increase their profits, and companies with lower tax rates are more willing to use stock options. Aboody and Kasznik [12] showed that if firms choose to reward shareholders by paying dividends, then this would reduce the value of stock options. However, the value of restricted stock is not affected by dividend payments. As a result, executives are more likely to prefer to adopt restricted stock plans and increase the dividend payout rate to meet shareholder expectations.

According to the relevant domestic research, there is little literature on the choice of equity incentive model in China, most of which discusses the innovation mechanism triggered by equity incentive, while some literature combines equity incentive with executive power to explore the reasons for choosing equity incentive model in Chinese enterprises [13–16]. Xiao, et al [17] and others argue that the more power executives have, the more they tend to choose restricted stock with more room for profit, and that restricted stock is a "welfare" incentive. Zong, et al [18] found that both models reduce the probability of executive sales/senior executive change in companies. Tian and Meng [19] indicated that both incentive models promote corporate innovation, but when the stock price is close to the exercise price, the punitive nature of restricted stock for executives affects the incentive to innovate, and stock options protect executives and motivate corporate innovation. Su and Alexiou [20] argued that the equity incentive model significantly affected the incentive effect, and that enhancing the incentive intensity of the stock option model could positively affect R&D expenditures. Concentration of equity can strengthen the extent of the effect of stock option model incentives on R&D

expenditures. Zhang, et al [21] found that executive compensation incentives, executive equity incentives and executive power incentives all positively affect the quality of corporate innovation; independent directors play a positive moderating role in the relationship between executive compensation incentives, executive equity incentives, executive power incentives and corporate innovation quality. On the whole, domestic studies are less likely to explore the causes of incentive model selection and do not delve into the inner spirit of executives inspired by different incentive models.

As a major component of market economic activities, entrepreneurs have the spirit to play a significant role in the business management activities of enterprises [22]. At present, the concept of entrepreneurship is rather vague in academic circles, and it is a difficult research task to classify the dimensions and measure entrepreneurship in a scientific way. Miler [23] and Covin [24] measure entrepreneurship from different perspectives, which include the entrepreneur's ability to innovate, compete, take risks, work together, and take initiative. Lumpkin and Dess [25] argue that entrepreneurship consists of five main elements: innovation, change, adventure, independence, and competitiveness. Covin [26] classified entrepreneurship in terms of psychological signs into four dimensions: innovation, adventure, collaboration, and effort. Li [27] pointed out that entrepreneurship, led by the spirit of innovation, was very broad and rich in content, and also included the spirit of competition and adventure as well as the sensitivity to the outside world, which could drive the steady economic growth of enterprises. It is clear from the literature that no matter how scholars define entrepreneurship, the core of the spirit remains the spirit of innovation and adventure. The equity incentive system closely links the interests of executives with those of the company. This benefit-sharing mechanism can stimulate the adventure spirit of executives and thus motivate them to explore new market opportunities and innovative products more actively in order to achieve the long-term development of the company and long-term returns for shareholders. Baker [28] stated that stock option incentive models motivate executives to manage greater risks and increase the market value of the firm. Murphy [29] showed that stock options could motivate executives to better manage firm risk and thus improved enterprise performance. Nienhaus [30] explored how equity incentives affect executives' adventure behavior. The study found that stock option plans can motivate executives to take more risks, which in turn promotes firm growth and development. The study also found that equity incentive plans could align the interests of executives and shareholders, which in turn could improve the performance and value of the firm. In addition, many studies have shown that equity incentives can stimulate innovation among executives. For example, Hall and Murphy's [31] study argued that stock options could stimulate executives' ability to create new products and services, thereby increasing the firm's market capitalization. Carpenter and Petersen [32] also showed that stock options could stimulate executives' innovative thinking and willingness to innovate, thus promoting firm innovation.

In the existing literature on equity incentives and enterprise performance, linear regression models, case methods, and two-party game models are mainly used for the study. This paper combines mathematical analysis and linear regression to study the relationship between the equity incentive model, entrepreneurship, and corporate performance. This changes the monotonicity of the ways in traditional research and verifies the relationship between the three from both theoretical derivation and empirical analysis, making the experimental results better and more reliable.

## 3 Theoretical analysis and research hypothesis

According to the Measures for the Administration of Equity Incentives for Listed Companies (for Trial Implementation), China's equity incentive program requires the clear establishment

of performance evaluation indicators as a condition for incentive recipients to receive equity. Therefore, incentive recipients need to put in some effort to achieve investment success and drive performance growth in order to meet the criteria for receiving equity. Different equity incentive models can lead to differences in core covenant settings, which in turn can lead to large differences in grant costs. On the one hand, the restricted stock incentive model stipulates that "the issue price shall not be less than 50% of the average price of the company's shares for the 20 trading days prior to the pricing reference date". Under the stock option model, it is stipulated that "the exercise price shall not be less than the par value of the stock and shall not be less than one of the average trading prices of the company's stock for the 20 trading days, 60 trading days or 120 trading days prior to the announcement of the draft equity incentive plan". This shows that the cost for incentive recipients to acquire restricted stock is lower than that of stock options. On the other hand, there are differences in the way different incentive models are granted. Under the stock option incentive model, the incentive recipient does not need to pay any money on the grant date and only needs to cancel the registered stock options without any loss if the assessment target is not completed; If the assessment is met, the listed company normally pays the incentive recipient directly the cash equivalent of the stock option to complete the award. Under the restricted stock model, the incentive recipient has to pay real money to buy the granted shares immediately on the grant date. If the evaluation target is not completed, the listed company repurchases the corresponding stock, but the repurchase funds obviously cannot offset the financing cost and opportunity cost of the incentive object; If the assessment is up to standard, the incentive object can benefit from selling the unlocked shares, but when the incentive recipient is a director, supervisor, or officer of the company, the percentage of shares sold will be limited. As a result, the incentive recipients of restricted stock face more constraints and take longer to realize the benefits. In summary, the differences between the two incentive models are shown below. (See Table 1)

Based on the above analysis, the hypothesis conditions proposed in this paper are shown below:

A1: If equity incentives stimulate adventure among executives, the average cost required to motivate executives to make risky investments is AC.

A2: The probability that the executive venture is successful and meets the performance target is $\beta$. The value of $\beta$ takes the range $(0.5,1)$. It is assumed that the executive maintains the original level of decision making and also achieves the assessment goal. If $\beta$ takes a value less than 0.5, the probability of meeting the test after the executive makes a venture capital investment is small, and the executive will not make a venture capital investment.

A3: The opening stock price is IP, the stock option grant price is IP, and the restricted stock grant price is $\alpha$IP, with $\alpha$ taking values in the range $(0.5,1)$. The cost (including capital cost and opportunity cost) for incentive recipients to purchase restricted stock is amortized to OC per share, and OC is much less than the stock grant price IP.

**Table 1. Diagram of differences in equity incentive models.**

| Differences | Restricted Stock | Stock Options |
|---|---|---|
| Grant price | In practice it is mostly granted at half price | Generally granted for parity |
| Grant Method | 1. A one-time purchase of the granted shares at the beginning of the period, subject to financing costs and opportunity costs. | 1.No need to buy shares at the beginning of the period, no need to assume the purchase capital. |
| | 2.After the performance target is reached, if the incentive targets are directors and supervisors, the stock realizations receive regulatory restrictions. | 2. After the performance target is met, the listed company generally pays the equivalent amount of cash for the difference, and the realization of shares is not restricted. |

**Table 2. Executive venture capital selection.**

| Equity Incentive Model | Make venture capital investments | Give up venture capital |
|---|---|---|
| Stock Options | Risky investments under stock options | Shedding risky investments under stock options |
| Restricted Stock | Risky investments under restricted stock | Shedding of risky investments under restricted stock |

A4: Before the implementation of the equity incentive plan, the probability that an executive chooses a risky project is λ. λ takes values in the range [0,1]. The cost of the original decision of the incentive recipient is (1-λ) AC. Due to the fact that executive salaries were not linked to performance before the implementation of the equity incentive plan, executives were prone to misuse of power and lazy behavior. If a company implements an equity incentive system, it will stimulate the adventure spirit of executives, which will lead them to make risky investments and increase the intensity of their work, which will increase the cost of their time and energy. Therefore, the more investment in risky projects, the lower the original decision-making cost.

A5: Assume that the stock price is only related to performance. When the investment is successful, the expected share price rises to HP, and when the investment fails to meet the appraisal target, the expected share price is LP, and LP < IP < HP.

Under the above conditions, we can obtain the payoff matrix for whether executives choose risky investments or not, as follows in Tables 2 and 3.

As can be seen from Table 3, the conditions for executives to choose the restricted stock incentive model are shown below:

(1) If venture capital is made:

$$\beta(HP - \alpha IP - AC - OC) + (1 - \beta)(-AC - OC) > \beta(HP - IP - AC) + (1 - \beta)(-AC) \quad (1)$$

This inequality is organized to give:

$$IP > OC/(1 - \alpha)\beta \quad (2)$$

(2) If no venture capital is made:

$$\alpha IP - (1 - \lambda)(AC - OC) > -(1 - \lambda)AC \quad (3)$$

This inequality is organized to give:

$$\alpha IP + (1 - \lambda)OC > 0 \quad (4)$$

From the above conditions, inequality (2)(4) holds.

The above comparison shows that E3>E1, E4>E2, and the restricted stock incentive model is more advantageous regardless of whether executives make the choice to make venture capital investments. This view is consistent with some findings in the existing literature, for example, Conyon and Murphy [33] compares CEO compensation and incentives in the US and the UK and found that restricted stock incentive plans could increase shareholder value and

**Table 3. Earnings matrix for executives.**

| Equity Incentive Model | Make venture capital investments | Give up venture capital |
|---|---|---|
| Stock Options | β(HP-IP-AC) +(1-β) (-AC) | -(1-λ) AC |
| Restricted Stock | β(HP-αIP-AC-OC) + (1-β) (-AC-OC) | αIP -(1-λ) (AC-OC) |

**Table 4. Matrix of benefits of considering executive adventure.**

| Equity Incentive Model | Choose Venture Capital | Maintain status quo |
|---|---|---|
| Stock Options | $\beta(HP\text{-}IP\text{-}AC) + (1\text{-}\beta)(\text{-}AC)$ | $-(1\text{-}\lambda) AC$ |
| Restricted Stock | $\beta(HP\text{-}\alpha IP\text{-}AC\text{-}OC) + (1\text{-}\beta)(\text{-}AC\text{-}OC)$ | $-(1\text{-}\lambda) AC\text{-}OC$ |

employee satisfaction and promoted long-term corporate growth. Kruse [34] argues that restricted stock incentive plans can increase firm productivity, profitability, and shareholder value. Because these incentive plans could improve employees' motivation and sense of belonging, promote the alignment of employees' interests with those of the firm, and thus improve enterprise performance. Ittner, et al. [35] analyzed data from more than 200 new economy firms and found that restricted stock incentive models could promote growth and innovation, increase employee motivation and creativity, and thus improve enterprise performance. The study also pointed out that the impact of restricted stock incentive plans on enterprise performance was related to the structure and implementation of the incentive plans. Based on the above theory and analysis, this paper proposes hypothesis H1:

H1: In companies that implemented equity incentive systems, the equity incentive model had a significant impact on enterprise performance. Compared to stock options, firms that implemented restricted stock incentive model performed better.

The purpose of implementing the equity incentive system in listed companies is to mitigate the principal-agent problem. The Relevant Income Matrix See Table 4 below.

The condition for executives to make venture decisions under the stock option model is:

$$\beta(HP - IP - AC) + (1 - \beta)(-AC) > -(1 - \lambda)AC \tag{5}$$

This inequality is organized to give:

$$\lambda < \beta(HP - IP)/AC \tag{6}$$

The condition for executives to make risk-based decisions under the restricted stock model is:

$$\beta(HP - \alpha IP - AC - OC) + (1 - \beta)(-AC - OC) > -(1 - \lambda)AC - OC \tag{7}$$

This inequality is organized to give:

$$\lambda < \beta(HP - \alpha IP)/AC \tag{8}$$

Combining the above inequalities (6)(8), we can derive the following results. When $\lambda < \beta$ (HP-IP)/AC, the incentive recipients will make risky investments in both models. When $\lambda \geq \beta$ (HP-$\alpha$IP)/AC, incentive recipients forgo risky investments in both models. When $\beta$(HP -IP) /AC$\leq \lambda < \beta$(HP-$\alpha$IP) /AC, the incentive recipients of restricted stock will make risky investments, while risky investments will be foregone under stock options. Therefore, the restricted stock incentive model is more likely to stimulate the adventure spirit of executives than stock options. In addition, from a theoretical point of view, since the amount of company stock held by executives is closely related to the performance of the company's stock, executives are more motivated to take more risks in order to earn higher returns. Adventure behavior by executives may include exploring new markets, launching new products, investing in R&D, etc. All of these behaviors are likely to promote innovation and growth in the company, thereby improving business performance. Since the restricted stock incentive model is characterized by the need for executives to retain their stock holdings for a certain time period, it encourages them

to work hard for the long-term benefit of the company. Compared to the stock option incentive model, the restricted stock incentive model places more emphasis on the long-term goals of the company by preventing executives from making decisions that are detrimental to the long-term growth of the company in pursuit of short-term stock prices. Steven and Bernadette [36] explored the impact of restricted stock incentive models on CEO turnover rate. Results found that restricted stock incentive plans reduce CEO turnover rate because it motivates CEO to focus more on the long-term interests and success of the company, thereby promoting growth and development. Babak, et al. [37] studied and analyzed the effect of restricted stock incentive model on the degree of adventure of the company. The results of the study suggested that the restricted stock incentive model could motivate executives to take more risks and reduce the financial risk of the company, as executives could not earn in case of poor performance of the company's stock. However, when an executive makes any decision, the prerequisite is the executive's authority. Finkelstein [38] showed that different dimensions of power structure in executive teams had a significant impact on executive behavior and organizational performance. This indicated that different forms and degrees of executive power affect executive decision making and behavior, which in turn affected organizational performance. Hambrick and Finkelstein's, et al. [39] study argued that executive discretion could affect organizational performance, but the effect was not always positive. Executive discretion may lead to increased executive adventure, but it may also lead to misconduct or excessive adventure by executives, which could harm organizational performance. Chen [40] argues that the restricted stock incentive model can motivate executives to take more risks and increase their focus on the value of the company's stock, thus promoting the growth of the company. Lyu and Chen [41] found equity financing had an enhanced mediating effect on the relevance between founder control and corporate performance. Summarizing the above perspectives from the literature, we see that firms with relatively more concentrated executive power have better performance. Therefore, hypothesis H2 is proposed in this paper.

H2: In firms implementing equity incentive systems, executive adventure plays a mediating effect between enterprise performance and equity incentive patterns, and performance is more pronounced in firms with more concentrated executive power.

In general, when companies undertake innovative projects such as R&D, the project risk level increases significantly. On the one hand, innovation projects require a continuous injection of large amounts of liquidity, and companies will pay a larger capital cost and opportunity cost. On the other hand, if such projects fail, it will be difficult to convert the invested capital, which will bring a large financial loss to the company. Under the above assumptions, it can be obtained that under the stock option incentive model, $\lambda < \beta$ (HP—IP)/ AC is satisfied and executives will make venture project investments, defined as B1. The risky decision of executives in the restricted stock model is conditional on $\lambda < \beta$(HP-$\alpha$IP) /AC, defined as B2. The analysis conducted from Table 3 above shows that the condition for executives to invest in venture projects under the restricted stock model is E3 > E4, i.e., $\lambda < [\beta$ (HP - $\alpha$IP) - $\alpha$IP] /AC, denoted as B3. Comparing B3 with B1 and B2, we can see that B3 requires a higher risk level for the investment. Under the restricted stock incentive model, the range of $\lambda$ corresponding to risky investment by executives is stricter, which can play a better incentive effect in innovative projects with lower investment success rate, i.e., the restricted stock incentive model can better stimulate the innovation spirit of executives. Besides, restricted stock incentive has a stronger sense of goal binding and risk taking for executives, which can better stimulate their sense of responsibility and belonging, and thus enhance their sense of innovation and innovation ability. In a restricted stock incentive, executives are required to achieve specific performance goals or company development goals within a certain period of time in order to receive

the corresponding stock award. Such incentives are more explicit and specific for executives, who need to constantly strive to innovate and improve their business models in order to reach their goals and be rewarded. In contrast, stock option incentives are more flexible, and executives are free to choose whether to exercise their options based on market changes, lacking mandatory constraints on targets. In an empirical study of Chinese listed companies, Li [42] concluded that restricted stock incentives can significantly improve corporate innovation and stimulate executive innovation, thus promoting corporate innovation and development. Hambrick [43] analyzed the impact of executives' personality traits, perceptions and experiences on corporate strategy and innovation from the perspective of upper management and concluded that restricted stock incentives can increase executives' willingness and behavior to innovate. Cai and Wei [44], through an empirical study of listed companies in the Chinese A-share market, found that restricted stock incentives can effectively stimulate executives' innovation awareness and innovative behavior. Yi, et al. [45] find that venture capital could significantly promote open innovation of enterprises. Based on the above analysis, the following hypotheses are proposed:

H3: In firms implementing equity incentive systems, executive innovation plays a mediating effect between enterprise performance and equity incentive patterns, and firms with higher R&D investment perform more favorably.

## 4 Research design

### 4.1 Sample selection and data sources

This paper uses the data of listed companies in China A-share market from 2010 to 2020 as the sample. Meanwhile, to ensure the accuracy and validity of the data, the sample data were screened as follows: (1) Excluding financial and insurance listed companies. (2) Exclude companies that terminate their listing. (3) Exclude listed companies that adopt hybrid model. (4) Exclude companies that have suspended or terminated the implementation of equity incentives. (5) Excluding listed companies with vacant main explanatory variables. The data sources of this paper are mainly the CSMAR and CHOICE databases. The missing data for this paper are firstly crawled by a Python big data crawler, and then the mean value of the sample period of this listed company is used as a substitute. Data screening is performed through Excel, and after Excel processing, 1872 observations are finally obtained, and empirical analysis is completed through Stata.

### 4.2 Definition of variables

**4.2.1. Explained variables.** Enterprise Performance (ROA). The indicators used in the study to measure enterprise performance mainly include accounting indicators and market indicators. Accounting indicators include return on equity (ROE) and return on total assets (ROA), while market indicators are mainly measured by earnings per share (EPS), Tobin's Q, price-to-earnings ratio (P/E), and market capitalization-to-book ratio (M/B) [46]. Foreign scholars mostly use market indicators to measure business performance in their research on enterprise performance. However, it is worthy of consideration that there is still a gap between the development of China's capital market and foreign countries, the effectiveness of the capital market is weak, and market indicators may not match the actual market situation, the rationality of using market indicators to measure enterprise performance is questionable. On the other hand, market indicators are usually linked to stock prices, but the frequency of stock price changes in our market and the weak market validity usually make it impossible to accurately measure enterprise performance using market indicators. At present, when scholars in

China study this issue, the indicators selected in most studies are still based on return on net assets (ROE) and return on total assets (ROA). Synthesizing previous studies and the requirements of the Management Measures of Equity Incentives for Listed Companies, this paper selects the return on average annual total assets (ROA) to measure enterprise performance and uses the return on average annual net assets (ROE) as the explanatory variable for the robustness test.

**4.2.2. Explanatory variables.** *4.2.2.1. Equity incentive model (MODE).* This paper selects listed companies that implement two incentive models, stock options and restricted stock, and sets dummy variables: the implementation of restricted stock incentive model takes the value of 1, and the stock option incentive model takes the value of 0.

*4.2.2.2. Adventurousness (HAZARD).* This paper draws on Xie, et al. [47] who selects three indicators of betting agreements, leverage, and foreign investment in new areas to measure the adventurousness of firms. The higher ratio of capital employed by the three is taken as 1 when compared with the same industry, otherwise it is 0.

*4.2.2.3. INNOVATION.* Innovation is the primary characteristic of entrepreneurship. On the one hand, invention patents are an important indicator of enterprise innovation, and on the other hand, a strong research team and sufficient investment in R&D are the keys to occupy the technological heights and obtain core patents. Therefore, this paper adopts the proportion of patent applications and R&D investment of enterprises in the past three years as indicators to measure innovation spirit, both of which are greater than the industry average taking the value of 1, otherwise 0.

*4.2.2.4. Executive Power (MODEPOWER).* Executive power is a more comprehensive concept that generally refers to its ability to suppress disagreement and the ability of executives to carry out their own wishes. The main reference in this paper is Finkelstein's [38] classification, which was also referenced by Quan and Wu, et al. [48], and a comprehensive analysis was conducted using principal component analysis on four dimensions of management's organizational structure power, ownership power, prestige power, and expert power to finally arrive at the management decision-making power variable. First, in measuring the power of management organizational structure, this paper selects a value of 1 when the chairman is also the general manager and 0 otherwise. Second, in the study of ownership power, ownership power is measured in terms of two dimensions: the percentage of executive shareholding as well as equity dispersion, and the ownership measure in this paper is mainly borrowed from Wang, et al. [49], which uses equity dispersion for measurement. It takes the value of 1 when the shareholding of the first largest shareholder is less than the sum of the shareholding of the second to tenth largest shareholders, and 0 otherwise. Third, in measuring prestige power this paper uses the age of the general manager as a measure of prestige power. It is generally believed that the older the age, the more experience the job has, the more social connections it has and the more prestige it has. The general manager's age is greater than the sample mean age takes the value of 1, otherwise it is 0. Fourth, expert power comes from the management's deep expertise and rich working experience in a certain field, mainly referring to the general manager's ability to coordinate the internal and external environment and handle unexpected events. This article draws on the approach of Quan, et al. [48]. The article selects the length of tenure of the general manager to measure, and the longer the tenure, the more experience he or she has in management. Then the more prestige he or she has accumulated in the enterprise, the more control he or she has over the enterprise. The number of years of service of the general manager is greater than the average age of the sample takes the value of 1, otherwise it is 0. If the value of management's organizational structure power, ownership power, prestige power and expert power is greater than 0.5 with 25% each, the paper considers management's decision making authoritative, then its and takes the value of 1, otherwise it is 0.

MODEPOWER is the interaction term between MODE and POWER, which indicates whether the executive power is more concentrated under the restricted stock incentive model implemented by listed companies.

*4.2.2.5. R&D investment (MODERDINVESTMENT).* In this paper, when considering R&D investment, we also take into account the listed enterprises' investment in human and material resources, and use the ratio of R&D expenses and R&D personnel, both of which are greater than the industry average to take the value of 1, otherwise take the value of 0. MODERDINVESTMENT is the interaction term between MODE and RDINVESTMENT, which indicates the R&D investment under the implementation of restricted stock incentive model in listed companies.

**4.2.3. Control variables.** The company's financial performance is also disturbed by other factors, such as company size, operating capacity, debt capacity, governance, and capital utilization. This paper chooses company size (SIZE), total asset sales/revenue rate (OPERATION), asset-liability ratio (LEV), enterprise donation (DONATIONAMOUNT), company qualification (FIRMAGE), institutional shareholding (INST), audit situation (BIG4), major shareholder capital occupation (OCCUPY), whether it is a state-owned enterprise (SOE) as control variables. The definition and interpretation of each variable are as follows, (see Table 5).

## 4.3 Model building

Impact of equity incentive models on enterprise performance:

$$ROA_{i,t} = \alpha_0 + \alpha_1 MODE_{i,t} + \varepsilon_{i,t} \tag{9}$$

$$ROA_{i,t} = \alpha_{10} + \alpha_{11} MODE_{i,t} + \alpha_{12} CONTROLS_{i,t} + \varepsilon_{i,t} \tag{10}$$

Mediating effects of executive adventure:

$$HAZARD_{i,t} = \alpha_{20} + \alpha_{21} MODE_{i,t} + \alpha_{22} CONTROLS_{i,t} + \varepsilon_{i,t} \tag{11}$$

$$ROA_{i,t} = \alpha_{30} + \alpha_{31} MODE_{i,t} + \alpha_{32} HAZARD_{i,t} + \alpha_{33} CONTROLS_{i,t} + \varepsilon_{i,t} \tag{12}$$

Impact of executive power:

$$ROA_{i,t} = \alpha_{40} + \alpha_{41} MODEPOWER_{i,t} + \alpha_{42} CONTROLS_{i,t} + \varepsilon_{i,t} \tag{13}$$

Mediating effects of executive innovativeness:

$$INNOVATION_{i,t} = \alpha_{50} + \alpha_{51} MODE_{i,t} + \alpha_{52} CONTROLS_{i,t} + \varepsilon_{i,t} \tag{14}$$

$$ROA_{i,t} = \alpha_{60} + \alpha_{61} MODE_{i,t} + \alpha_{62} INNOVATION_{i,t} + \alpha_{63} CONTROLS_{i,t} + \varepsilon_{i,t} \tag{15}$$

Impact of R&D investment:

$$ROA_{i,t} = \alpha_{70} + \alpha_{71} MODERDINVESTMENT_{i,t} + \alpha_{72} CONTROLS_{i,t} + \varepsilon_{i,t} \tag{16}$$

# 5 Empirical results and analysis

## 5.1 Descriptive statistical analysis

According to the descriptive statistics of the variables in Table 6, it can be seen that among the A-share listed companies implementing equity incentives: the mean value of ROA is 6.95%,

**Table 5. Variable definitions and descriptions.**

| Category | Name | | Symbols | Definition | references |
|---|---|---|---|---|---|
| **Explained variables** | **Return on Total Assets** | | **ROA** | **Net income / Average balance of total assets** | **[46]** |
| Explanatory variables | Motivation Model | | MODE | The restricted stock value is 1, and the stock option value is 0. | [10, 12] |
| | Decision-making power | Whether the chairman and the general manager are concurrently appointed | MODEPOWER | The value is 1 when the power is more concentrated, otherwise it is 0 | [48, 49] |
| | | Management shareholding ratio | | | |
| | | Age | | | |
| | | Term of office | | | |
| | R&D investment | R&D expenses as a percentage | MODERDINVESTMENT | The percentage of both weights is greater than the industry average of 1, otherwise it is 0 | [20] |
| | | R&D staff ratio | | | |
| | Spirit of adventure | Betting against | HAZARD | Capital employed/Total assets is greater than the industry average of 1, otherwise it is 0 | [38, 40, 47] |
| | | M&A restructuring | | | |
| | | New areas of foreign investment | | | |
| | Spirit of Innovation | Patent Application Rate | INNOVATION | Both weights are greater than the industry average value of 1, otherwise it is 0 | [42, 43, 45] |
| | | R&D cost investment ratio | | | |
| Control variables | Company size | | SIZE | Ln (Total assets) | [4–6, 13–18, 46] |
| | Total asset turnover rate | | OPERATION | (Opening assets + Closing assets)/2 | |
| | Asset-liability ratio | | LEV | Liabilities at the end of the period/Total assets at the end of the period | |
| | Enterprise donation | | DONATIONAMOUNT | Ln (Enterprises donate assets) | |
| | Company qualification | | FIRMAGE | Company establishment period | |
| | Institutional shareholding | | INST | Institutional ownership | |
| | Audit situation | | BIG4 | The value of the four major audit institutions is 1, otherwise 0. | |
| | Major shareholder capital occupation | | OCCUPY | Percentage of capital employed by major shareholders | |
| | Whether it is a state-owned enterprise | | SOE | The value of state-owned enterprises is 1, otherwise 0. | |

the standard deviation is 0.054, and the median value is 6.44%, which indicates that the enterprise performance of listed companies implementing equity incentive system is better. The mean value of MODE is 0.7473 with a median bias toward 1. It proves that listed companies in China are now preferring to use the restricted stock incentive model; the mean value of HAZARD is 35.63% with a standard deviation of 0.479 and a median bias toward 1. This shows that there are significant differences in the risky asset allocation of different listed companies and the business strategies chosen by executives vary. The mean value of MODEPOWER is 0.9444, the standard deviation is 0.229, and the median is 1, which proves that the power of listed companies implementing equity incentive system is relatively concentrated. Meanwhile, the mean value of INNOVATION is higher than 0.5, which also shows that the listed companies implementing the equity incentive system focus on conducting R&D projects with good input-output ratio, reflecting the spirit of innovation. The differences between industries can lead to different evaluation criteria for innovation input-output efficiency, as also shown by the standard deviation of 0.5823 for INNOVATION in Table 7. In addition, the higher INNOVATION but not MODERDINVESTMENT proves that the firms implementing the restricted stock incentive model focus on innovation efficiency and do not continuously invest more in innovation. Since enterprise performance is tied to executive earnings, the efficiency of executive decision making is improved.

**Table 6. Results of descriptive statistics of variables.**

| VarName | Obs | Mean | SD | Min | Median | Max |
|---|---|---|---|---|---|---|
| ROA | 1872 | 0.0695 | 0.054 | -0.1261 | 0.0644 | 0.2411 |
| MODE | 1872 | 0.7473 | 0.435 | 0.0000 | 1.0000 | 1.0000 |
| HAZARD | 1872 | 0.3563 | 0.479 | 0.0000 | 0.0000 | 1.0000 |
| MODEPOWER | 1872 | 0.9444 | 0.229 | 0.0000 | 1.0000 | 1.0000 |
| INNOVATION | 1872 | 0.5823 | 0.493 | 0.0000 | 1.0000 | 1.0000 |
| MODERDINVESTMENT | 1872 | 0.2895 | 0.454 | 0.0000 | 0.0000 | 1.0000 |
| LEV | 1872 | 0.3780 | 0.185 | 0.0530 | 0.3711 | 0.8325 |
| SIZE | 1872 | 20.8616 | 1.605 | 17.7273 | 20.7076 | 25.5418 |
| OPERATION | 1872 | 0.7000 | 0.431 | 0.1341 | 0.5971 | 2.7824 |
| DONATIONAMOUNT | 1872 | 1.14e+04 | 4.85e+05 | 0.0000 | 0.0000 | 2.10e+07 |
| FIRMAGE | 1872 | 2.7840 | 0.357 | 1.0986 | 2.8332 | 3.6636 |
| INST | 1872 | 0.3363 | 0.233 | 0.0000 | 0.3176 | 0.9379 |
| BIG4 | 1872 | 0.0459 | 0.209 | 0.0000 | 0.0000 | 1.0000 |
| OCCUPY | 1872 | 0.0151 | 0.025 | 0.0000 | 0.0079 | 0.3446 |
| SOE | 1872 | 0.0994 | 0.299 | 0.0000 | 0.0000 | 1.0000 |

Note: ***, **, * indicate significant at the 1%, 5%, 10% levels; same as below.

Specifically, it appears that among the A-share listed companies implementing equity incentives: the mean value of *ROA* is 6.95% with a standard deviation of 0.054. The *ROA* of 1820 out of 1876 samples was positive, which leads to the conclusion that most listed enterprises implementing equity incentives have better effects on profits and the distribution of enterprise performance (see Fig 1).

**Table 7. Regression results of equity incentive model.**

| | ROA | | ROA | |
|---|---|---|---|---|
| | Coefficient | t | Coefficient | t |
| MODE | 0.0091** | (1.9780) | 0.0106** | (2.5324) |
| LEV | | | -0.2090*** | (-7.3774) |
| SIZE | | | 0.0202*** | (3.8501) |
| OPERATION | | | 0.0845*** | (6.8267) |
| DONATIONAMOUNT | | | 0.0000** | (2.2056) |
| FIRMAGE | | | -0.0265 | (-1.1171) |
| INST | | | 0.0278** | (2.5562) |
| BIG4 | | | -0.0027 | (-0.3503) |
| OCCUPY | | | 0.0117 | (0.1404) |
| SOE | | | -0.0096 | (-0.5674) |
| _cons | 0.0901*** | (10.7183) | -0.2520** | (-2.1796) |
| TE | Yes | | Yes | |
| FE | Yes | | Yes | |
| N | 1872 | | 1872 | |
| R2 | 0.0736 | | 0.2915 | |
| Adj. R2 | 0.0681 | | 0.2839 | |

Note

***, **, * indicate significant at the 1%, 5%, 10% levels.

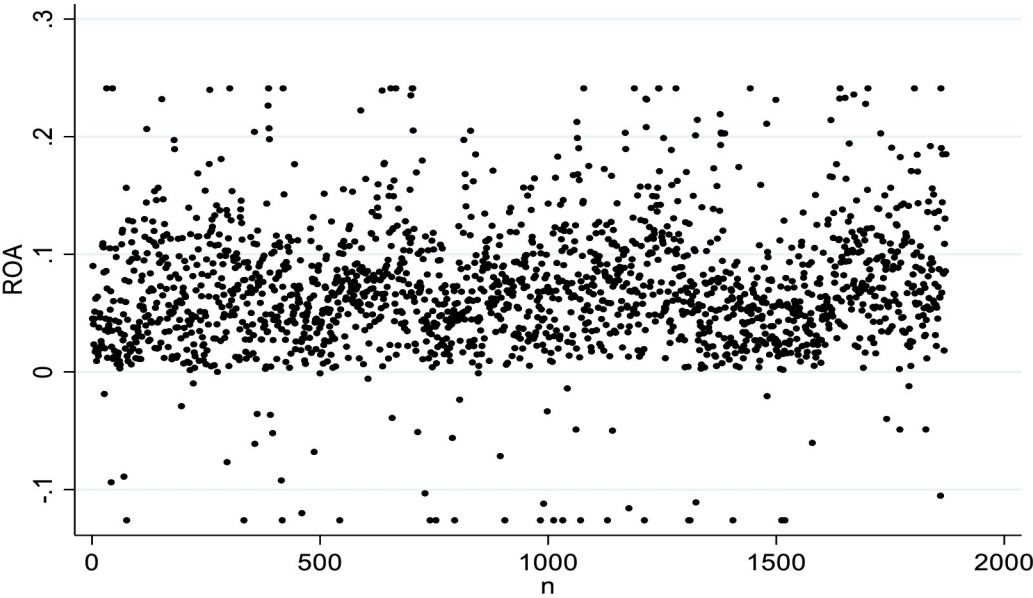

**Fig 1. Enterprise performance distribution map.**

## 5.2 Regression analysis

In this paper, we use panel data to regress the data of listed companies. The regression results of model (9)(10) are reported in Table 7.

Column (1) addresses the impact of equity incentive model on firms. Restricted stock incentive model has a positive effect on enterprise performance at the 5% level of significance, conditional on the inclusion of no control variables. Column (2) still has a significant positive effect with the addition of control variables and a large increase in adjusted $R^2$. The regression results demonstrate that the use of restricted stock incentive model by listed companies motivates executives while creating constraints for executives to strive to maximize enterprise performance. Hypothesis H1 holds.

If the above assumptions H2 and H3 hold, the conduction mechanism shown in Fig 2 is satisfied. The MODE and HAZARD coefficients in columns (1) and (2) in Table 8 are both significantly positive at the 1% level. This means that executive adventure plays a mediating role between equity incentive model and enterprise performance, and a mediating mechanism exists for restricted stock incentives to stimulate executive adventure and thus enhance enterprise performance. Thus, the implementation of restricted stock incentive model in listed companies can reduce the lazy behavior of executives, and stimulate the adventure spirit of executives, motivate them to obtain the maximum income and promote the improvement of enterprise performance.

The coefficient of MODEPOWER in column (1) of Table 9 is also significant and positive, which implies that in firms with relatively more concentrated executive power among those

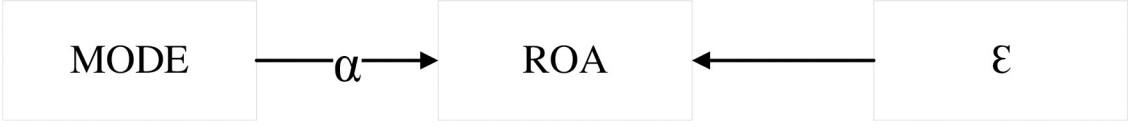

**Fig 2. Intermediary mechanism conduction diagram.**

**Table 8. Mediating effects of executive adventurism.**

| | HAZARD | | ROA | |
|---|---|---|---|---|
| | Coefficient | t | Coefficient | t |
| MODE | 0.4814*** | (11.0713) | 0.0019 | (0.4159) |
| LEV | -0.9472*** | (-3.7158) | -0.1919*** | (-6.8775) |
| SIZE | 0.1345*** | (3.1252) | 0.0178*** | (3.4302) |
| OPERATION | 0.2097*** | (2.5922) | 0.0808*** | (6.5515) |
| DONATIONAMOUNT | -0.0000 | (-0.9130) | 0.0000** | (2.3966) |
| FIRMAGE | -0.3763* | (-1.9444) | -0.0197 | (-0.8458) |
| INST | 0.1704 | (1.6259) | 0.0247** | (2.2774) |
| BIG4 | -0.2394 | (-1.1690) | 0.0016 | (0.2083) |
| OCCUPY | 0.7513 | (1.2231) | -0.0018 | (-0.0205) |
| SOE | 0.0256 | (0.1879) | -0.0101 | (-0.6495) |
| HAZARD | | | 0.0180*** | (4.3495) |
| _cons | -1.5497* | (-1.6527) | -0.2241** | (-1.9919) |
| TE | Yes | | Yes | |
| FE | Yes | | Yes | |
| N | 1872 | | 1872 | |
| R2 | 0.2598 | | 0.3157 | |
| Adj. R2 | 0.2518 | | 0.3080 | |

Note

***, **, * indicate significant at the 1%, 5%, 10% levels.

**Table 9. Interaction term regression results.**

| | ROA | | ROA | |
|---|---|---|---|---|
| | Coefficient | t | Coefficient | t |
| LEV | -0.2084*** | (-7.2192) | -0.2073*** | (-7.3235) |
| SIZE | 0.0202*** | (3.8000) | 0.0201*** | (3.8273) |
| OPERATION | 0.0829*** | (6.5020) | 0.0849*** | (6.6894) |
| DONATIONAMOUNT | 0.0000*** | (2.7324) | 0.0000* | (1.8955) |
| FIRMAGE | -0.0234 | (-0.9695) | -0.0216 | (-0.9080) |
| INST | 0.0285*** | (2.5919) | 0.0261** | (2.3861) |
| BIG4 | -0.0029 | (-0.4472) | -0.0048 | (-0.6240) |
| OCCUPY | -0.0135 | (-0.1642) | -0.0035 | (-0.0400) |
| SOE | -0.0096 | (-0.5685) | -0.0122 | (-0.7064) |
| MODEPOWER | 0.0128** | (2.1501) | | |
| MODERDINVESTMENT | | | 0.0128*** | (3.0678) |
| _cons | -0.2624** | (-2.1947) | -0.2562** | (-2.1796) |
| TE | Yes | | Yes | |
| FE | Yes | | Yes | |
| N | 1872 | | 1872 | |
| R2 | 0.2859 | | 0.2949 | |
| Adj. R2 | 0.2782 | | 0.2873 | |

Note

***, **, * indicate significant at the 1%, 5%, 10% levels.

**Table 10. Mediating effects of executive innovation spirit.**

|  | INNOVATION | | ROA | |
|---|---|---|---|---|
|  | Coefficient | t | Coefficient | t |
| MODE | 0.1360** | (2.2108) | 0.0094** | (2.2536) |
| LEV | -0.5583 | (-1.6428) | -0.2043*** | (-7.3551) |
| SIZE | 0.0619 | (1.1332) | 0.0197*** | (3.7682) |
| OPERATION | 0.0380 | (0.3253) | 0.0842*** | (6.8580) |
| DONATIONAMOUNT | 0.0000*** | (5.4118) | 0.0000 | (1.4873) |
| FIRMAGE | 0.0740 | (0.2703) | -0.0272 | (-1.1309) |
| INST | -0.0869 | (-0.6409) | 0.0285*** | (2.6661) |
| BIG4 | -0.6920*** | (-2.6401) | 0.0031 | (0.3343) |
| OCCUPY | 0.9315 | (0.7819) | 0.0039 | (0.0489) |
| SOE | -0.1181 | (-0.5812) | -0.0086 | (-0.4851) |
| INNOVATION |  |  | 0.0084*** | (2.7738) |
| _cons | -1.0770 | (-0.8748) | -0.2430** | (-2.0893) |
| TE | Yes | | Yes | |
| FE | Yes | | Yes | |
| N | 1872 | | 1872 | |
| R2 | 0.0746 | | 0.3019 | |
| Adj. R2 | 0.0646 | | 0.2940 | |

Note

***, **, * indicate significant at the 1%, 5%, 10% levels.

implementing the restricted stock incentive model, executives are able to make full use of their decision-making power and strive to achieve their performance goals. Sun, et al. [50] study pointed out that the relative concentration of power within the executive team can improve the financial performance of the firm, and this concentration can improve strategic positioning and resource allocation. Wang, et al. [51] showed that the relative concentration of power within the executive team can improve the operational performance of the firm, and this concentration can improve the efficiency of decision making and execution. The above studies all point out that public companies with more centralized executive power are more efficient in executive decision making and provide prerequisites for executives to make risky decisions. The test results are consistent with Hambrick, et al. [39], Babak, et al. [37], and Chen [40]. Hypothesis H2 was verified.

MODE and INNOVATION are significantly positive in columns (1) and (2) of Table 10, MODE is significant at the 5% level and INNOVATION is significant at the 1% level. That means entrepreneurial innovation plays a mediating effect between equity incentive model and firm performance. Mechanistic tests show that the equity incentive model of a firm stimulates executive innovation and thus has an impact on enterprise performance. In a public company with restricted stock incentive model, executives have actually paid the cost, but at the same time, executives will get a high amount of income once the performance is achieved. Executives often start by investing in R&D projects because of the high risk and high return of innovative projects compared to normal investment projects, and the restricted stock gives the executives ties and stimulates the executives' innovation. MODERDINVESTMENT in column (8) is the investment in R&D projects by listed firms under the restricted stock incentive model. Its regression coefficient is significantly positive at the 1% level, indicating that executives in firms implementing the restricted stock incentive model invest more in R&D in order to meet performance targets. Yu, et al. [52] found through their empirical study from 2010–

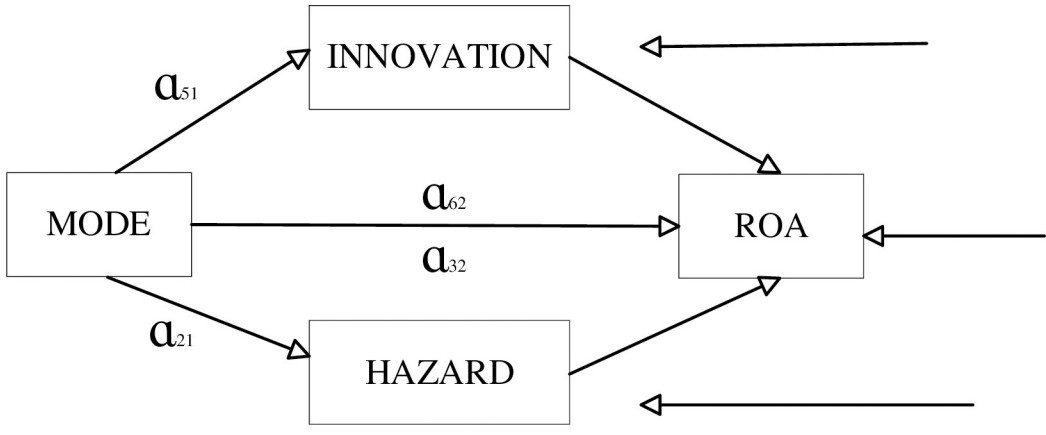

**Fig 3. Map of the mediating role of entrepreneurship.**

2016 that R&D investment and innovation have a positive effect on enterprise performance with the effect of R&D investment on enterprise performance being influenced by firm size, form of ownership and industry type. Akram and Rizwan [53] empirical study based on 2010–2019 found that R&D investment and innovation have a positive effect on enterprise performance as well as the effect of R&D investment on enterprise performance is influenced by the type of industry. The above experimental results are in agreement with such literature. Hypothesis H3 was verified.

In summary, executive adventure and innovation play a mediating role between equity incentive models and firm performance. The impact mechanism is that the implementation of restricted stock incentive model in listed companies stimulates the adventure and innovation spirit of executives, which in turn has an impact on corporate performance. The impact mechanism is illustrated in Fig 3 below.

### 5.3 Robustness tests

In order to ensure the accuracy and consistency of the above findings, the paper applied the substitution variable method to test the above results. The standard error calculation formula proposed by Sobel [54] is used to test the mediating effect:

$$S_{ab} = \sqrt{b^2 s_a^2 + a^2 s_b^2} \tag{17}$$

We use the *ab* product as the sampling distribution, which is shown as the magnitude of the mediating effect. The test of whether the *ab* product is significant is a direct test of the mediating effect.

According to the regression results of Table 11 and Table 12, the regression results are still significantly positive after replacing the variables. p-values of Aroian in the Sobel test were less than 0.05, all passed the test of mediating effects, and the significance levels of regression coefficients were basically consistent with the above.

### 5.4 Endogenous test

In this paper, the import/export balance (lnbalance), the growth index of other funds for fixed asset investment (lninvest1), the growth index of net disposable property income per resident (lnincom), and the growth index of fixed asset investment (lninvest) in the same period are used as instrumental variables in addressing the indigeneity problem. Also considering that

**Table 11. Substitution of explained variables.**

|  | ROE01 | | ROE03 | | ROE05 | |
|---|---|---|---|---|---|---|
|  | **Coefficient** | **t** | **Coefficient** | **t** | **Coefficient** | **t** |
| MODE | 0.0186** | (2.1883) |  |  |  |  |
| LEV | -0.4211*** | (-3.1288) | -0.4202*** | (-3.1140) | -0.4182*** | (-3.1155) |
| SIZE | 0.0607*** | (2.8719) | 0.0605*** | (2.8599) | 0.0604*** | (2.8673) |
| OPERATION | 0.1873*** | (4.0719) | 0.1844*** | (3.9856) | 0.1881*** | (4.0629) |
| DONATIONAMOUNT | 0.0000 | (1.5507) | 0.0000** | (1.9734) | 0.0000 | (1.3129) |
| FIRMAGE | 0.0126 | (0.2459) | 0.0182 | (0.3511) | 0.0213 | (0.4086) |
| INST | 0.0302 | (1.4795) | 0.0315 | (1.5380) | 0.0271 | (1.3198) |
| BIG4 | 0.0044 | (0.3951) | 0.0042 | (0.4371) | 0.0008 | (0.0710) |
| OCCUPY | -0.0274 | (-0.2047) | -0.0725 | (-0.5376) | -0.0542 | (-0.3884) |
| SOE | -0.0284 | (-0.8441) | -0.0282 | (-0.8724) | -0.0329 | (-0.9863) |
| MODEPOWER |  |  | 0.0234** | (2.3687) |  |  |
| MODERDINVESTMENT |  |  |  |  | 0.0230*** | (2.7774) |
| _cons | -1.0917** | (-2.3664) | -1.1112** | (-2.3830) | -1.0994** | (-2.3709) |
| TE | Yes | | Yes | | Yes | |
| FE | Yes | | Yes | | Yes | |
| N | 1872 | | 1872 | | 1872 | |
| R2 | 0.2506 | | 0.2469 | | 0.2534 | |
| Adj. R2 | 0.2425 | | 0.2388 | | 0.2453 | |

Note

***, **, * indicate significant at the 1%, 5%, 10% levels.

the change of any economic factor itself has a certain inertia, the results of the previous period usually have an impact on the results of the later period, and there is a lag effect on the enterprise performance of each listed company. Therefore, this paper uses systematic generalized moment estimation (GMM) to estimate a dynamic panel data model for robustness analysis of potential indigeneity problems. The regression results passed the stability test, and the coefficient symbols were consistent with the above. The P values of Hansen test were 0.846,0.991 and 0.995, respectively. The GMM test passes, then the model does not have indigeneity problems. (See Table 13)

# 6 Discussion and conclusion

## 6.1 Discussion

A void in current literature pertains to examining the equity incentive model in conjunction with entrepreneurship and its impact on firm performance. Existing research has predominantly centered on scrutinizing executive decision-making behavior through lenses such as compensation structure, shareholding ratios, and perspectives on risk-taking. Addressing this gap, Gan and Yang [55] employed dynamic programming methods to construct a principal-agent model, presenting numerical results and conducting quantitative analyses of optimal contracts. The findings indicated that, in comparison to pure equity compensation, contingent compensation served as an effective mechanism in mitigating executives' inclination towards risk-taking decisions. This prompted executives to consider the broader societal welfare and emphasize a long-term development focus for the enterprise. Coles, et al. [56] posited that equity functions as a call option contingent upon the firm's asset value, with higher risk elevating the equity's worth. Consequently, executives, incentivized by equity, are inclined towards

**Table 12. Sobel test for mediation effect.**

| | ROE02 | | ROE04 | |
|---|---|---|---|---|
| | Coefficient | t | Coefficient | t |
| MODE | 0.0035 | (0.3651) | 0.0167* | (1.9343) |
| LEV | -0.3914*** | (-2.8932) | -0.4134*** | (-3.1082) |
| SIZE | 0.0564*** | (2.6506) | 0.0598*** | (2.8545) |
| OPERATION | 0.1808*** | (3.8727) | 0.1868*** | (4.1046) |
| DONATIONAMOUNT | 0.0000* | (1.6788) | 0.0000 | (1.0719) |
| FIRMAGE | 0.0244 | (0.4818) | 0.0116 | (0.2235) |
| INST | 0.0249 | (1.1855) | 0.0314 | (1.5707) |
| BIG4 | 0.0119 | (1.0048) | 0.0139 | (0.9654) |
| OCCUPY | -0.0510 | (-0.3630) | -0.0402 | (-0.3120) |
| SOE | -0.0292 | (-0.9136) | -0.0267 | (-0.7597) |
| HAZARD | 0.0313*** | (3.5205) | | |
| MODEPOWER | | | | |
| INNOVATION | | | 0.0138** | (2.2811) |
| MODERDINVESTMENT | | | | |
| _cons | -1.0431** | (-2.2621) | -1.0768** | (-2.3440) |
| TE | Yes | | Yes | |
| FE | Yes | | Yes | |
| N | 1872 | | 1872 | |
| R2 | 0.2671 | | 0.2569 | |
| Adj. R2 | 0.2587 | | 0.2485 | |
| Sobel Z | 3.211 | | 1.536 | |
| Sobel Z-p | 0.001 | | 0.000 | |
| Aroian Z | 3.200 | | 1.461 | |
| Aroian Z-p | 0.001 | | 0.000 | |
| Goodman Z | 3.223 | | 1.623 | |
| Goodman Z-p | 0.001 | | 0.000 | |
| Intermediary effect as a percentage | 0.813 | | 0.101 | |

Note

***, **, * indicate significant at the 1%, 5%, 10% levels.

engaging in ventures with increased risk. This perspective supports Christopher S. Armstrong's, et al. [57] assertions, highlighting a positive correlation between equity compensation holdings and executive proclivity for risk. Building on this, Zhang's, et al. [58] research revealed an inverted U-shaped trend in enterprise innovation investment concerning the proportion of equity compensation in executive remuneration. Executive risk-taking gauged through Delta and Vega indices, emerged as a mediating factor influencing corporate innovation paths shaped by compensation structures. Heterogeneity analysis underscored the more pronounced innovation incentive effect of higher executive equity compensation in non-state-owned firms and those in highly marketized regions. Wang, et al. [59] dissected the mechanisms through which equity incentives operate, identifying the "risk-taking" and "golden handcuffs" effects. Equity incentives impacted firms' investment in exploratory innovation primarily through the "risk-taking" effect, while their influence on utilizing innovation was attributed to the "golden handcuffs" effect. In contrast, the mediating effect of the risk-taking component on utilizing investment was deemed non-significant. Hao and Zhang's [60] categorization of executive incentives into explicit and implicit incentives, explicitly focusing on

**Table 13. GMM Endogenous test.**

| | ROA | | ROA | | ROA | |
|---|---|---|---|---|---|---|
| | Coefficient | t | Coefficient | t | Coefficient | t |
| MODE | 0.009** | (0.004) | | | | |
| LEV | -0.168*** | (0.041) | -0.307*** | (0.060) | -0.280*** | (0.045) |
| SIZE | 0.005 | (0.006) | 0.011* | (0.006) | 0.014*** | (0.005) |
| OPERATION | 0.024 | (0.024) | 0.011 | (0.021) | 0.014 | (0.021) |
| DONATIONAMOUNT | -0.000 | (0.000) | -0.000 | (0.000) | -0.000 | (0.000) |
| FIRMAGE | -0.011 | (0.039) | 0.037 | (0.030) | 0.023 | (0.029) |
| INST | 0.019 | (0.024) | 0.042* | (0.025) | 0.034 | (0.024) |
| BIG4 | 0.007 | (0.041) | 0.009 | (0.025) | 0.003 | (0.008) |
| OCCUPY | -0.021 | (0.205) | -0.318 | (0.267) | -0.086 | (0.165) |
| SOE | 0.002 | (0.027) | -0.004 | (0.029) | -0.017 | (0.035) |
| MODEHAZARD | | | 0.038** | (0.018) | | |
| MODEINNOVATION | | | | | 0.017** | (0.008) |
| TE | Yes | | Yes | | Yes | |
| FE | Yes | | Yes | | Yes | |
| N | 1872 | | 1872 | | 1861 | |
| AR(1)p | 0.106 | | 0.077 | | 0.128 | |
| AR(2)p | 0.344 | | 0.363 | | 0.351 | |
| Hansenp | 0.846 | | 0.991 | | 0.995 | |

Note

***, **, * indicate significant at the 1%, 5%, 10% levels.

linear incentives like equity, revealed a masking effect on risk-taking. Explicit incentives, such as equity, were found to exert a more substantial influence on R&D investment through the mediating effect of assuming significant risks, unlike implicit incentives like promotions. This study further delineates executive equity incentives into two models—restricted stock incentive and stock option incentive—reflecting prevalent practices in the Chinese market. Mathematical analysis comparing financing costs and equity realization constraints underlines that relative to the stock option incentive model, the restricted stock incentive model effectively stimulates executives' risk-taking disposition in decision-making, steering them towards innovative projects with lower success rates. Ultimately, this model, geared towards maximizing executive interests, engenders a culture of risk-taking and innovation, impacting overall enterprise performance.

Moreover, existing research endeavors to probe the pivotal role of entrepreneurship in influencing enterprise performance. Zhang, et al. [61] demonstrated that entrepreneurship is vital to fostering enterprise performance and economic growth. Notably, entrepreneurship has a more pronounced impact on economic growth in the central and northeastern regions than in the eastern and western regions. Using principal components analysis, Jiang and Zhang [62] constructed entrepreneurship and enterprise performance indicators. The outcomes underscored that entrepreneurship significantly enhances enterprise performance through innovation and entrepreneurial endeavors. Peters and Waterman's [54] research further supported the notion that entrepreneurship contributes to heightened financial performance in enterprises. Mao, et al. [63], employing a Structural Equation Model (SEM), positioned entrepreneurship as a central factor in the interplay between organizational learning, innovation, and firm performance. Results indicated no direct effect between entrepreneurship and enterprise performance, with enterprise performance indirectly influenced by the mediating

variables of organizational learning and innovation. Chen and Wei's [64] questionnaire survey, covering 112 enterprises in China's manufacturing, service, and information industries, led to developing a structural equation model through data analysis. The findings revealed that entrepreneurship influences firm performance through the mediating variable of organizational learning. Additionally, environmental uncertainty affects firm performance through entrepreneurship and organizational learning. Despite the considerable attention given to executive incentives, entrepreneurship, and firm performance in corporate governance literature, there needs to be more scholarly work integrating these three facets. Hence, this study introduces entrepreneurship as a mediating variable to scrutinize the interplay between the equity incentive model, entrepreneurship, and firm performance. The first hypothesis (H1) posits that the restricted stock incentive model positively impacts firm performance, aligning with the findings of Abody [12] and Tong [65]. Building upon this foundation, hypotheses H2 and H3 assert that the entrepreneurial spirit of executives serves as the transmission mechanism between the equity incentive model and enterprise performance. Results affirm that the restricted stock incentive model positively influences enterprise performance by invigorating executives' adventurous and innovative spirit. Further analysis demonstrates that enterprises with more centralized executive decision-making power and higher research and development (R&D) investment exhibit superior performance. This suggests that the restricted stock incentive model does not compel executives to engage in blind, risky investments. Instead, executives make informed decisions grounded in thorough market research, strategic planning, and investment decisions, continuously stimulating their motivation and positively impacting corporate performance. This study contributes valuable insights to understanding entrepreneurship as a mechanism and transmission path in corporate dynamics.

## 6.2 Conclusion

This paper explores the relationship between equity incentive model, entrepreneurship and enterprise performance at the theoretical level. On the basis of this paper, empirical analysis is also conducted and the following conclusions are drawn: among the firms implementing equity incentive system, restricted stock incentive model has a positive impact on enterprise performance. The adventure and innovation spirit of executives play a mediating role between the equity incentive model and enterprise performance, firms with more concentrated executive power perform more prominently, and firms with more R&D investment perform better under the restricted stock incentive model.

## 7. Limitation and implication

This study, focusing on listed companies and executives, explores the diverse characteristics of two incentive models. It comprehensively considers the stakes of executives under different incentive structures and, based on this understanding, endeavors to stimulate the adventurous and innovative spirit of executives. This, in turn, aims to mitigate complacency and opportunistic behaviors in executive decision-making, ultimately fostering the growth of enterprise performance. However, it is essential to note that China's capital market started relatively late. While current market efficiency has improved significantly, it has yet to reach a semi-strong level of effectiveness. This may impact the overall efficacy of the equity incentive system.

Given the ongoing transition of the equity incentive system in the Chinese market, this paper confines its analysis to the present market context. Consequently, a more in-depth exploration of the relationship between incentive models, entrepreneurship, and enterprise performance, considering the unique characteristics of the Chinese market, remains a crucial avenue for further investigation in subsequent studies [66].

Furthermore, with the deepening of enterprise digitalization, future research could employ advanced methods such as artificial neural networks to scrutinize these factors and refine the understanding of how different incentive models and characteristics impact firm performance. Additionally, delving into non-material incentives, such as positions, power, and honors, could provide valuable insights into executive motivation and governance practices.

## Author Contributions

**Conceptualization:** Jinrong Ma, Hongbo Wang.

**Data curation:** Jinrong Ma.

**Formal analysis:** Jinrong Ma.

**Funding acquisition:** Hongbo Wang.

**Project administration:** Hongbo Wang.

**Resources:** Jinrong Ma.

**Supervision:** Hongbo Wang.

**Writing – original draft:** Jinrong Ma.

**Writing – review & editing:** Hongbo Wang.

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
