## [Decision Letter · Decision Letter 0]

29 Nov 2023

PONE-D-23-16055Research on the Impact of Equity Incentive Model on Enterprise Performance: A Mediating Effect Analysis Based on Executive EntrepreneurshipPLOS ONE

Dear Dr. Ma,

Thank you for submitting your manuscript to PLOS ONE. After careful consideration, we feel that it has merit but does not fully meet PLOS ONE’s publication criteria as it currently stands. Therefore, we invite you to submit a revised version of the manuscript that addresses the points raised during the review process.

We look forward to receiving your revised manuscript.

Kind regards,

Valentina Diana Rusu, PhD

Academic Editor

PLOS ONE

Journal Requirements:

2 Thank you for stating the following in the Acknowledgments Section of your manuscript:

"This research was funded by Natural Science Foundation of China [No. 71771112], and Project of Liaoning Provincial Federation Social Science Circles of China [No. L20BGL047，L16BJY011]."

"This research was funded by Natural Science Foundation of China [No. 71771112], and Project of Liaoning Provincial Federation Social Science Circles of China [No. L20BGL047，L16BJY011]. "

"This research was funded by Natural Science Foundation of China [No. 71771112], and Project of Liaoning Provincial Federation Social Science Circles of China [No. L20BGL047，L16BJY011]. "    

Reviewers' comments:

Reviewer's Responses to Questions

**Comments to the Author**

1. Is the manuscript technically sound, and do the data support the conclusions?

Reviewer #1: Yes

Reviewer #2: No

2. Has the statistical analysis been performed appropriately and rigorously? 

Reviewer #1: Yes

Reviewer #2: No

3. Have the authors made all data underlying the findings in their manuscript fully available?

Reviewer #1: Yes

Reviewer #2: No

4. Is the manuscript presented in an intelligible fashion and written in standard English?

Reviewer #1: Yes

Reviewer #2: No

5. Review Comments to the Author

Reviewer #1: Dear Editor,

I have completed reviewing this article (PONE-D-23-16055) which needs major revision.

Title: Research on the Impact of Equity Incentive Model on Enterprise Performance: A Mediating Effect Analysis Based on Executive Entrepreneurship

Suggestion: Major revision

This article studies the selection of listed enterprises regarding the equity incentive model. The authors followed the logical idea that the equity incentive system can stimulate the adventure and innovation spirit of executives and thus have a positive impact on enterprise performance. They employed a combination of numerical and mathematical analysis to investigate the selection of equity incentive models. The findings suggest that the choice of restricted stock incentive model for listed enterprises implementing equity incentive system has a significant positive effect on enterprise performance. The Mechanistic tests revealed that both executive adventure and innovation play a mediating role in the influence of restricted stock incentive models on enterprise performance. When a company implements the restricted stock incentive model, executives will maximize their own interests as starting point and be willing to take more risky investments. This behavior will stimulate the adventure spirit of executives, which will have a positive impact on enterprise performance, and is more prominent in companies with more concentrated power of executives. It is a very meaningful paper which gives the public some guidances when starting to make some policies on equity incentive in enterprise performance. It is well-organized and designed. My overall feeling towards this manuscript is positive. Analysis is also appropriate and seems properly implemented. However, there are some major issues that need to be addressed.

1. Abstract. What is the limitation of the existing studies? The authors need to explore the originality and contribution of the study in this section.

2. The introduction section lacks research gaps presentation. What are research gaps in existing literature? The authors should explicitly state in this section and then draw your contributions smoothly.

3. The authors should provide a paragraph at the end of the Introduction section to present the article organization.

4. The literature section is weak; How the existing studies related to your methods? These issues should be stated clearly in this section.

5. The data source should be stated more clearly, and how the authors pre-process the null data?

6. The discussion section is missed; the authors should focus on your results, or the recommendations should be drawn from your results directly.

7. The limitations or future research directions should be presented in a new subsection.

8. please updated some newest references.

Reviewer #2: Dear Authors and Editor,

Firstly, thank you for submitting your paper “Research on the Impact of Equity Incentive Model on Enterprise Performance: A Mediating Effect Analysis Based on Executive Entrepreneurship” (# PONE-D-23-16055) to PLOS ONE. After reading, I should point out these issues. The following is from my comments.

1. The authors should check the grammar and tense throughout the whole manuscript.

2. The authors should check the language throughout the whole manuscript.

3. The authors should check the keywords and keep the order of the keywords based on the capital letter “A-Z” and 4-5 preferred.

4. In the introduction, the authors should pay more attention to the recent and important literature. The following is upon the topic “Equity Incentive, Enterprise Performance, Executive Entrepreneurship” that the authors can cite, I suggest that

Does venture capital help to promote open innovation practice? Evidence from China

https://doi.org/10.1108/EJIM-03-2021-0161

Effect of Founder Control on Equity Financing and Corporate Performance-Based on Moderation of Radical Strategy

https://doi.org/10.1177/21582440221085013

Military Experience and Individual Entrepreneurship—Imprinting Theory Perspective: Empirical Evidence From China

https://doi.org/10.1177/21582440231159866

5.The authors should add value and process of the study in the section.

6. The authors pay more attention to the definitions of variables, which is very important, adding the references into Table 5. Besides, the authors also add the fig. of research model.

7.The authors should add more details in the methodology section and how to collect data.

8. The authors can change the expression of fig. 1.

9. The authors add the meaning of *, **, ** in table 8 and table 9, table 10, table 11 and table 12 and table 13.

10. The authors should add more to the “Discussion section”, meanwhile, comparing the result with previous ones. The authors should write the discussion section, however, it is missing.

11. The authors should add conclusion section, however, it is also missing.

12. The authors should check the important and recent ones.

13. The authors should check all the references’ formats based on Plos One’s style. Besides, some lack pages. Besides, the authors should pay more attention to the text-in ciatations through the whole manuscript again and check strictly one by one.

To summarize, I should choose “Major Revision” and welcome the revised version in the future.

All the best

Nov 09 ,2023

6. PLOS authors have the option to publish the peer review history of their article (what does this mean?). If published, this will include your full peer review and any attached files.

Reviewer #1: No

Reviewer #2: **Yes: **Bei Lyu

---

## [Author Response · Author response to Decision Letter 0]

22 Jan 2024

Dear Editors and Reviewers: 

Thank you for your letter and for the reviewers’ comments concerning our manuscript entitled “Research on the Impact of Equity Incentive Model on Enterprise Performance: A Mediating Effect Analysis Based on Executive Entrepreneurship” (Manuscript Number: PONE-D-23-16055). We sincerely thank the editor and all reviewers for their valuable feedback that we have used to improve the quality of our manuscript. These comments are all valuable and helpful for improving our article. According to your nice suggestions, we have made extensive corrections to our previous manuscript. The detailed corrections are listed below. The revised parts of the article have been marked up using the Word Microsoft Track Changes tool. For the convenience of reviewers, we have uploaded two versions: manuscript-Revised Manuscript with Track Changes (Word Microsoft Track Changes) and manuscript- Manuscript.

Answers to Review Report (Manuscript Number: PONE-D-23-16055):

1. Reviewer #1

Q1: Abstract. What is the limitation of the existing studies? The authors need to explore the originality and contribution of the study in this section.

Response: 

Thanks for your suggestion. Additional information about the limitations, originality, and contributions of the study has been provided in the abstract.

Limitations:

See the manuscript – Manuscript: lines 29-32. it is marked in yellow.

or the manuscript – Revised Manuscript with Track Changes (Word Microsoft Track Changes): lines 52-55. it is marked in yellow.

Originality:

See the manuscript – Manuscript: lines 10-16. it is marked in yellow.

or the manuscript – Revised Manuscript with Track Changes (Word Microsoft Track Changes): lines 33-39. it is marked in yellow.

Contribution:

See the manuscript – Manuscript: lines 32-34. it is marked in yellow.

or the manuscript – Revised Manuscript with Track Changes (Word Microsoft Track Changes): lines 55-57. it is marked in yellow.

Q2: The introduction section lacks research gaps presentation. What are research gaps in existing literature? The authors should explicitly state in this section and then draw your contributions smoothly.

Response: 

We believe that your suggestions are very useful to us. We have described the gaps between this study and the existing literature in the introduction section.

See the manuscript – Manuscript: lines 78-85. it is marked in yellow.

or the manuscript – Revised Manuscript with Track Changes (Word Microsoft Track Changes): lines 181-188. it is marked in yellow.

Q3: The authors should provide a paragraph at the end of the Introduction section to present the article organization.

Response: 

Thanks for the suggestion. We have added an introduction to the organization of the article at the end of the introduction.

See the manuscript – Manuscript: lines 98-110. it is marked in yellow.

or the manuscript – Revised Manuscript with Track Changes (Word Microsoft Track Changes): lines 201-213. it is marked in yellow.

Q4: The literature section is weak; How the existing studies related to your methods? These issues should be stated clearly in this section.

Response: 

We think your comments are very helpful. Based on the comments you made, we think it is very necessary to add the Literature Review section, so we added the Literature Review section. And we have summarized the existing research methods for this issue and explained the research methodology of this paper. For details, see 2. Literature Review. 

See the manuscript – Manuscript: lines 112-195. Due to the excessive length of the section, it is not marked in yellow.

or the manuscript – Revised Manuscript with Track Changes (Word Microsoft Track Changes): lines 215-298. Due to the excessive length of the section, it is not marked in yellow.

Q5: The data source should be stated more clearly, and how the authors pre-process the null data?

Response: 

Thanks for the heads up, We have added the treatment of null data in the data sources section.

See the manuscript – Manuscript: lines 395-402. it is marked in yellow.

or the manuscript – Revised Manuscript with Track Changes (Word Microsoft Track Changes): lines 504-513. it is marked in yellow.

Q6: The discussion section is missed; the authors should focus on your results, or the recommendations should be drawn from your results directly.

Response: 

Thanks to your suggestion, we have revised both the title and the content of the last section of the article, which contains results, discussion and conclusion. The section on results and recommendations focuses on 6.2 Conclusion, which first summarizes the results of the paper and then makes relevant recommendations based on the results.

Results section:

See the manuscript – Manuscript: lines 729-734. it is marked in yellow.

or the manuscript – Revised Manuscript with Track Changes (Word Microsoft Track Changes): lines 853-858. it is marked in yellow.

Recommendation section:

See the manuscript – Manuscript: lines 739-742. it is marked in yellow.

or the manuscript – Revised Manuscript with Track Changes (Word Microsoft Track Changes): lines 863-866. it is marked in yellow.

Q7: The limitations or future research directions should be presented in a new subsection.

Response: 

Thanks to your suggestions, we have added comparisons and discussions with the existing literature and have proposed future research directions for this paper in a separate subsection.

See the manuscript – Manuscript: lines 743-748. it is marked in yellow.

or the manuscript – Revised Manuscript with Track Changes (Word Microsoft Track Changes): lines 867-872. it is marked in yellow.

Q8: Please updated some newest references.

Response: 

We have updated the references based on the suggestions you have made. See References 11,16,20,23,29,30,42,46,52,55,64,65 and 66 for details.

2.Reviewer #2: 

Q1: The authors should check the grammar and tense throughout the whole manuscript. 

Response: 

Thanks to your valuable comments, we have checked the manuscript for grammar and tenses and have touched up key parts of the paper. Details are given in Word Microsoft Track Changes. 

Q2: The authors should check the language throughout the whole manuscript.

Response: 

Thank you for your input for us, we have checked the language of the manuscript and made changes to some of the statements. See Word Microsoft Track Changes for details.

Q3: The authors should check the keywords and keep the order of the keywords based on the capital letter “A-Z” and 4-5 preferred.

Response:

Thanks for the suggestion. We kept the keywords to 4 or 5 and sorted them in A-Z order. 

See the manuscript – Manuscript: lines 35-36, it is marked in yellow.

or the manuscript – Revised Manuscript with Track Changes (Word Microsoft Track Changes): lines 58-59, it is marked in yellow.

Q4: In the introduction, the authors should pay more attention to the recent and important literature. The following is upon the topic “Equity Incentive, Enterprise Performance, Executive Entrepreneurship” that the authors can cite, I suggest that

Does venture capital help to promote open innovation practice? Evidence from China. https://doi.org/10.1108/EJIM-03-2021-0161

Effect of Founder Control on Equity Financing and Corporate Performance-Based on Moderation of Radical Strategy. https://doi.org/10.1177/21582440221085013

Military Experience and Individual Entrepreneurship—Imprinting Theory Perspective: Empirical Evidence From China. https://doi.org/10.1177/21582440231159866

Response: 

Thanks to your valuable comments, we have cited the three references you recommended in the manuscript and reworked the introductory section. The cited references can be found in references 42,43,55. 

See the manuscript – Manuscript: lines 341-342,383-384 and 739-742. it is marked in yellow.

or the manuscript – Revised Manuscript with Track Changes (Word Microsoft Track Changes): lines 445-446,492-493 and 863-866. it is marked in yellow.

Q5: The authors should add value and process of the study in the section. 

Response: 

Based on the revisions you have suggested, we have included the research value and process in the introduction section. 

Value：

See the manuscript – Manuscript: lines 86-97.

or the manuscript – Revised Manuscript with Track Changes (Word Microsoft Track Changes): 165-180.

Process：

See the manuscript – Manuscript: lines 98-110. it is marked in yellow.

or the manuscript – Revised Manuscript with Track Changes (Word Microsoft Track Changes): lines 201-213. it is marked in yellow.

Q6: The authors should add more details in the methodology section and how to collect data.

Response:

Following your suggestions, we have added to the text how to collect data and the processing of vacancy data. 

See the manuscript – Manuscript: lines 397-402, it is marked in yellow.

or the manuscript – Revised Manuscript with Track Changes (Word Microsoft Track Changes): lines 508-513, it is marked in yellow.

Q7: The authors can change the expression of fig. 1.

Response: 

Thank you for your revision, we have changed the presentation of Figure 1 in the text.

Q8: The authors add the meaning of *, **, ** in table 8 and table 9, table 10, table 11 and table 12 and table 13. Response: 

Thanks to your suggestion, we have added *, **, ** to Tables 8 and 9, 10, 11 and 12 and 13.

Q9: The authors should add more to the “Discussion section”, meanwhile, comparing the result with previous ones. The authors should write the discussion section, however, it is missing. 

Response: 

In response to your suggestion, we have added the “Discussion section” to Part 6 of the article and compared the results of this article with previous results.

See the manuscript – Manuscript: 647-727. Due to the excessive length of the section, it is not marked in yellow.

or the manuscript – Revised Manuscript with Track Changes (Word Microsoft Track Changes): 771-851. Due to the excessive length of the section, it is not marked in yellow.

Q10: The authors should add conclusion section, however, it is also missing. 

Response: 

Thank you for your advice, which is reflected in the conclusion section of the article.

See the manuscript – Manuscript: 729-734.

or the manuscript – Revised Manuscript with Track Changes (Word Microsoft Track Changes): 853-858.

Q11: The authors should check the important and recent ones. 

Response: 

We have updated the references based on the suggestions you have made. See References 11,16,20,23,29,30,42,46,52,55,64,65 and 66 for details.

Q12: The authors should check all the references’ formats based on Plos One’s style. Besides, some lack pages. Besides, the authors should pay more attention to the text-in ciatations through the whole manuscript again and check strictly one by one. 

Response: 

Thanks to your reminder, we have rechecked the formatting of all references according to Plos One style and have checked the citation of this article.

Thank you for your precious comments and advice. Those comments are all valuable and very helpful for revising and improving our paper. We have revised the manuscript accordingly, and our point-by-point responses are presented above.

Yours sincerely, 

Jinrong Ma

Address: Liaoning University of Science and Technology, Anshan City, China

Tel: +86-15638976977

E-mail: 15638976977@163.com

---

## [Decision Letter · Decision Letter 1]

31 Jan 2024

PONE-D-23-16055R1Research on the Impact of Equity Incentive Model on Enterprise Performance: A Mediating Effect Analysis Based on Executive EntrepreneurshipPLOS ONE

Dear Dr. Wang,

Thank you for submitting your manuscript to PLOS ONE. After careful consideration, we feel that it has merit but does not fully meet PLOS ONE’s publication criteria as it currently stands. Therefore, we invite you to submit a revised version of the manuscript that addresses the points raised during the review process.

**ACADEMIC EDITOR: Please take into account the recommendations made by reviewer 2, and change your paper accordingly.**

We look forward to receiving your revised manuscript.

Kind regards,

Valentina Diana Rusu, PhD

Academic Editor

PLOS ONE

Journal Requirements:

Reviewers' comments:

Reviewer's Responses to Questions

**Comments to the Author**

1. If the authors have adequately addressed your comments raised in a previous round of review and you feel that this manuscript is now acceptable for publication, you may indicate that here to bypass the “Comments to the Author” section, enter your conflict of interest statement in the “Confidential to Editor” section, and submit your "Accept" recommendation.

Reviewer #1: (No Response)

Reviewer #2: (No Response)

2. Is the manuscript technically sound, and do the data support the conclusions?

Reviewer #1: (No Response)

Reviewer #2: Yes

3. Has the statistical analysis been performed appropriately and rigorously? 

Reviewer #1: (No Response)

Reviewer #2: Yes

4. Have the authors made all data underlying the findings in their manuscript fully available?

Reviewer #1: (No Response)

Reviewer #2: No

5. Is the manuscript presented in an intelligible fashion and written in standard English?

Reviewer #1: (No Response)

Reviewer #2: No

6. Review Comments to the Author

Reviewer #1: I have reviewed this manuscript again, I think this manuscript could be considered for publication in present form.

Reviewer #2: Dear Authors and Editor,

Firstly, thank you for submitting your paper “Research on the Impact of Equity Incentive Model on Enterprise Performance: A Mediating Effect Analysis Based on Executive Entrepreneurship” (# PONE-D-23-16055R1) to PLOS ONE again. After reading, I should point out these issues. The following is from my comments.

1.The authors should reduce the number of the abstract, specially the content of results, and “By exploring this issue in depth, this paper aims to comprehensively evaluate the operation of China’s equity incentive system and provide a reference for listed companies to formulate the core covenants of the equity incentive system” should be the value of the manuscript.

2.The keywords should not be bold, so the authors should adjust them.

3.The authors should check the format of text-in citations, for example, it should be (Ma, 2024), not full name. So the authors should check all of them strictly.

4. In the introduction, the authors should pay more attention to the recent and important literature. Specially the year of 2024, the following is upon the topic “Entrepreneurship” that the authors can cite, I suggest that

Yi, R., Liu, S.S., & Lyu, B. A bibliometric and visualization analysis of Artisan entrepreneurship. Technology Analysis & Strategic Management, 2023; early access.

DOI: 10.1080/09537325.2023.2290152

5.The authors should uniform the format of all tables.

6.The authors should add the section of limitation and implication.

7.The authors should check all the references’ formats based on Plos One’s style. Besides, some lack pages. Besides, the authors should pay more attention to the text-in citations through the whole manuscript again and check strictly one by one.

8. The authors should make all the figs clearer.

To summarize, I should choose “Revision” and welcome the revised version in the future.

All the best

Jan 30,2024

7. PLOS authors have the option to publish the peer review history of their article (what does this mean?). If published, this will include your full peer review and any attached files.

Reviewer #1: No

Reviewer #2: **Yes: **Assoc. Prof. Dr. Bei Lyu

---

## [Author Response · Author response to Decision Letter 1]

16 Feb 2024

Dear Editors and Reviewers: 

Thank you for your letter and for the reviewers’ comments concerning our manuscript entitled “Research on the Impact of Equity Incentive Model on Enterprise Performance: A Mediating Effect Analysis Based on Executive Entrepreneurship” (Manuscript Number: PONE-D-23-16055R1). We sincerely thank the editor and all reviewers for their valuable feedback that we have used to improve the quality of our manuscript. These comments are all valuable and helpful for improving our article. According to your nice suggestions, we have made extensive corrections to our previous manuscript. The detailed corrections are listed below. The revised parts of the article have been marked up using the Word Microsoft Track Changes tool. For the convenience of reviewers, we have uploaded two versions: manuscript-Revised Manuscript with Track Changes (Word Microsoft Track Changes) and manuscript- Manuscript.

Answers to Review Report (Manuscript Number: PONE-D-23-16055R1):

Reviewer #2: 

Q1: The authors should reduce the number of the abstract, specially the content of results, and “By exploring this issue in depth, this paper aims to comprehensively evaluate the operation of China’s equity incentive system and provide a reference for listed companies to formulate the core covenants of the equity incentive system” should be the value of the manuscript. 

Response: 

Thanks to your valuable comments, We have reduced the conclusion part and modified the 'By exploring this issue in depth, this paper aims to comprehensively evaluate the operation of China’s equity incentive system and provide a reference for listed companies to formulate the core covenants of the equity incentive system ' part. Details are given in Manuscript: lines 29-31.

Q2: The keywords should not be bold, so the authors should adjust them.

Response: 

Thank you for your input for us, we have adjusted them. See Manuscript for details. Details are given in Manuscript: lines 32-33.

Q3: The authors should check the format of text-in citations, for example, it should be (Ma, 2024), not full name. So the authors should check all of them strictly.

Response:

Thanks for the suggestion. We re-examine the format of the citations in the text and modify them one by one. Details are given in Manuscript.

Q4: In the introduction, the authors should pay more attention to the recent and important literature. Specially the year of 2024, the following is upon the topic “Entrepreneurship” that the authors can cite, I suggest that Yi, R., Liu, S.S., & Lyu, B. A bibliometric and visualization analysis of Artisan entrepreneurship. Technology Analysis & Strategic Management, 2023; early access.DOI: 10.1080/09537325.2023.2290152

Response: 

Thanks to your valuable comments, we have cited the reference you recommended in the manuscript. The cited references can be found in reference 22.

Q5: The authors should uniform the format of all tables. 

Response: 

Based on the revisions you have suggested, We unify the format of all tables.

Q6: The authors should add the section of limitation and implication.

Response:

Following your suggestions, we add the 7th part: limitation and implication. Details are given in Manuscript: lines 733-752.

Q7: The authors should check all the references’ formats based on Plos One’s style. Besides, some lack pages. Besides, the authors should pay more attention to the text-in citations through the whole manuscript again and check strictly one by one.

Response: 

Thank you for your revision, we checked the format of all references and modified the corresponding references in the text.

Q8: The authors should make all the figs clearer. 

Response: 

Thanks to your suggestion, we have modified some of the fugs to make them look clearer.

Thank you for your precious comments and advice. Those comments are all valuable and very helpful for revising and improving our paper. We have revised the manuscript accordingly, and our point-by-point responses are presented above.

Yours sincerely, 

Jinrong Ma

Address: Liaoning University of Science and Technology, Anshan City, China

Tel: +86-15638976977

E-mail: 15638976977@163.com

---

## [Decision Letter · Decision Letter 2]

6 Mar 2024

Research on the Impact of Equity Incentive Model on Enterprise Performance: A Mediating Effect Analysis Based on Executive Entrepreneurship

PONE-D-23-16055R2

Dear Dr. Wang,

We’re pleased to inform you that your manuscript has been judged scientifically suitable for publication and will be formally accepted for publication once it meets all outstanding technical requirements.

Kind regards,

Valentina Diana Rusu, PhD

Academic Editor

PLOS ONE

Additional Editor Comments (optional):

Reviewers' comments:

Reviewer's Responses to Questions

**Comments to the Author**

1. If the authors have adequately addressed your comments raised in a previous round of review and you feel that this manuscript is now acceptable for publication, you may indicate that here to bypass the “Comments to the Author” section, enter your conflict of interest statement in the “Confidential to Editor” section, and submit your "Accept" recommendation.

Reviewer #2: All comments have been addressed

2. Is the manuscript technically sound, and do the data support the conclusions?

Reviewer #2: Yes

3. Has the statistical analysis been performed appropriately and rigorously? 

Reviewer #2: Yes

4. Have the authors made all data underlying the findings in their manuscript fully available?

Reviewer #2: Yes

5. Is the manuscript presented in an intelligible fashion and written in standard English?

Reviewer #2: Yes

6. Review Comments to the Author

Reviewer #2: Dear editor,

Based on the revised version, the authors already addressed these comments, I choose "Acceptance".

7. PLOS authors have the option to publish the peer review history of their article (what does this mean?). If published, this will include your full peer review and any attached files.

Reviewer #2: No

---

## [Editor Report · Acceptance letter]

27 Mar 2024

PONE-D-23-16055R2 

PLOS ONE

Dear Dr. Wang, 

I'm pleased to inform you that your manuscript has been deemed suitable for publication in PLOS ONE. Congratulations! Your manuscript is now being handed over to our production team.

Kind regards, 

on behalf of

Dr. Valentina Diana Rusu 

Academic Editor

PLOS ONE